# Using Immunoliposomes as Carriers to Enhance the Therapeutic Effectiveness of Macamide N-3-Methoxybenzyl-Linoleamide

**DOI:** 10.3390/neurolint17030038

**Published:** 2025-03-03

**Authors:** Karin J. Vera-López, María Aranzamendi-Zenteno, Gonzalo Davila-Del-Carpio, Rita Nieto-Montesinos

**Affiliations:** Facultad de Ciencias Farmacéuticas, Bioquímicas y Biotecnológicas, Universidad Católica de Santa María, Urb. San José s/n, Umacollo, Arequipa 04000, Peru; kvera@ucsm.edu.pe (K.J.V.-L.); marea1342@gmail.com (M.A.-Z.); gdavilad@ucsm.edu.pe (G.D.-D.-C.)

**Keywords:** nanomedicine liposomes, epilepsy, N-3-methoxybenzyl-linoleamide, F(ab′)_2_, OX26

## Abstract

Background/Objectives: Epilepsy is one of the most common chronic neurological disorders, characterized by alterations in neuronal electrical activity that result in recurrent seizures and involuntary body movements. Anticonvulsants are the primary treatment for this condition, helping patients improve their quality of life. However, the development of new drugs with fewer side effects and greater economic accessibility remains a key focus in nanomedicine. Macamides, secondary metabolites derived from Maca (*Lepidium meyenii*), represent a promising class of novel drugs with diverse therapeutic applications, particularly in the treatment of neurological disorders. Methods: In this study, we optimized the potential of the macamide N-3-methoxybenzyl-linoleamide (3-MBL) as an anticonvulsant agent through its encapsulation in PEGylated liposomes conjugated with OX26 F(ab′)_2_ fragments. Results: These immunoliposomes exhibited a size of 120.52 ± 9.46 nm and a zeta potential of −8.57 ± 0.80 mV. Furthermore, in vivo tests using a pilocarpine-induced status epilepticus model revealed that the immunoliposomes provided greater efficacy against epileptic seizures compared to the free form of N-3-methoxybenzyl-linoleamide at the same dose. Notably, the observed anticonvulsant effect was comparable to that of carbamazepine, a traditional FDA-approved antiepileptic drug. Conclusions: This pioneering work employs liposomal nanocarriers to deliver macamides to the brain, aiming to set a new standard for the use of modified liposomes in anticonvulsant epilepsy treatment.

## 1. Introduction

According to the study carried out by the Global Burden of Disease on the lead disorders affecting the nervous system [1], epilepsy ranks seventh worldwide among the neurological diseases with the greatest impact on disability-adjusted life years (DALYs). Epilepsy currently affects more than 50 million people worldwide and is considered the most common chronic non-contagious neurological disease [2,3]. Moreover, some studies point to a possible connection between the mechanisms that trigger epilepsy and neurodegenerative disorders [4,5,6]. Although the prevalence of active epilepsy is estimated to be from 4 to 10 people per 1000, it is estimated that around 80% of patients with epilepsy live in low- and middle-income countries [2,7]. The stigmatization of the disease, combined with ignorance, malpractice, and poorly applied or incomplete treatments, represents a serious public health and economic problem in these countries [2,7,8].

Recurrent episodic attacks and epileptic seizures characterize epilepsy due to alterations in neuronal electrical activity [2,9]. Although studies on the mechanisms that cause this disease have increased in the last decade, the causes of its pathogenesis have yet to be entirely determined [10]. According to the International League Against Epilepsy (ILAET), epilepsy etiologies are complex (currently, more than 15 different seizure types and 30 epilepsy syndromes are recognized [11,12]). Still, they can be grouped into immune, metabolic, structural, infectious, genetic, or unknown etiologies [9]. For example, specific epileptic syndromes have been associated with the progressive thinning of the cortex, loss of brain volume, and neuronal death [4,13,14].

At present, the approach for treating epilepsy focuses solely on alleviating symptoms and suppressing seizures through antiseizure medications (ASMs), which work by blocking sodium and calcium channels, enhancing GABAergic inhibition, and reducing excessive excitatory amino acid transmission [15,16]. However, these medications can have significant neurotoxic side effects (headache, blurred vision, nausea, fatigue, drowsiness, dizziness, and incoordination, among others), particularly in toddlers [12,17]. Various medications have been designed for the treatment of epilepsy, such as phenytoin, carbamazepine, valproic acid, cannabidiol, fenfluramine, and cenobamate [18,19]. However, antiseizure drugs (ASDs) are the most commonly prescribed, even though they do not necessarily have any action on the pathophysiological mechanisms of this disease [15,20]. These drugs have various therapeutic targetsm including the following: the blockade of voltage-gated sodium channels, potentiation of inhibitory neurotransmission, blockade of excitatory neurotransmission, modulation of neurotransmitter release, mTORC1 signaling, mechanisms in nonepileptic conditions, and the endocannabinoid system (ECS) [15,21].

Among the latest generation drugs that are aimed at modulating the ECS are macamides [22,23,24]. Extracted from the root of Maca (*Lepidium meyenii*), these secondary metabolites have shown neuroprotective effects [25,26,27,28]. In a recent article, we have demonstrated that two synthetic macamides had anticonvulsant effects in male Sprague Dawley rat models [29]. The macamides used for this purpose were N-3-methoxybenzyl-oleamide (3-MBO) and N-3-methoxybenzyl-linoleamide (3-MBL). Our in vivo results indicated that both compounds had similar anticonvulsant effects to diazepam during status epilepticus induced by pilocarpine, thereby improving survival rates. However, 3-MBL might be more potent than 3-MBO. Experimental evidence suggests that the therapeutic action of 3-MBL macamide may result from their ability to inhibit FAAH’s hydrolyzing action on endocannabinoids [22,27].

In this regard, Wu et al. evaluated the in vitro inhibitory activity of 19 macamides against the FAAH enzyme, including 3-MBL [30]. Their results showed that 3-MBL had a median Inhibitory Concentration (IC_50_) of 10.3 ± 1.3 μM, which was higher than the two standard FAAH inhibitors used as a control—OL-135 (IC_50_ = 0.033 μM) and PF-750 (IC_50_ = 1.09 μM). These authors suggest that the presence of the 3-methoxy substituent in the N-benzyl group helps the macamide-FAAH binding through hydrogen bond interactions with the Thr236 residue located in the cytosolic port of the enzyme. Similar results were achieved by Almukadi et al. (IC_50_ = 16 μM), who concluded that 3-MBL is an irreversible or slowly reversible inhibitor [31]. Moreover, Yu et al. have reported that 3-MBL improved cell viability, decreased LDH release and intracellular ROS generation, inhibited the reduction of MMP, and lowered the Bax/Bcl-2 ratio [27].

Despite the therapeutic properties exhibited by macamide 3-MBL, the development of drugs targeting the central nervous system (CNS) is challenging and requires increased treatment adherence to improve effectiveness [32]. Numerous ways have been explored for delivering drugs to the brain. These systems include nanoparticles, viral vectors, bio-conjugated vehicles, and vesicles, among others [32,33,34,35,36]. Recently, vesicular systems have garnered particular interest due to their ability to accommodate both hydrophobic and hydrophilic substances and their adaptable structures. Furthermore, the various transporters and receptors present on the Blood–Brain Barrier (BBB) offer the potential to employ biomolecules for delivering substances to the brain [32]. The most commonly used vesicles for drug transport are transfersomes, exosomes, virosomes, niosomes, and liposomes [32].

Discovered in the 1960s, liposomes are widely used for drug delivery to the brain for biological and technological benefits [37,38,39]. Constructed from nano- and micro-sized lipid bilayers with an aquos core, they can transport hydrophobic and hydrophilic agents. One advantage of liposomes is that their structure can be improved by adding macromolecules to optimize their diffusion in the bloodstream and their specific administration to the brain, making them less toxic and biocompatible [32,35,40]. For example, targeted liposomes can enhance the accumulation of liposomal drugs in the brain. This involves attaching targeting moieties, such as monoclonal antibodies (mAbs) or their fragments, like F(ab′)_2_, to polyethylene glycol (PEG) chains (Figure 1a) [41]. These immunoliposomes utilize biochemical transport systems at the BBB [41,42]. A notable mAb for this purpose is OX26, which targets Brain Capillary Endothelial Cells (BCECs) by binding to Transferrin Receptors (TfR) via receptor-mediated transcytosis without using the transferrin binding site (Figure 1b) [42,43,44].

In this study, we aimed to optimize the low therapeutic doses of N-3-methoxybenzyl-linoleamide to enhance its neuroprotective properties using drug delivery systems. Hence, N-3-methoxybenzyl-linoleamide-loaded PEGylated OX26 F(ab′)_2_ immunoliposomes were prepared, physicochemically characterized, and tested in vivo on a pilocarpine-induced status epilepticus model. The results demonstrated that N-3-methoxybenzyl-linoleamide-loaded PEGylated OX26 F(ab′)_2_ immunoliposomes exhibited greater anticonvulsant effects compared to the free form of N-3-methoxybenzyl-linoleamide, while being comparable to those of carbamazepine.

## 2. Methods and Materials

### 2.1. Materials

Synthetic N-3-methoxybenzyl-linoleamide (3-MBL) was synthesized in the Organic Chemistry Laboratory at MCPHS University [30]. 1,2-dimyristoyl-*sn*-glycero-3-phosphocholine (DMPC), MPEG(2000)-Distearoyl phosphatidylethanolamine (ammonium salt) (DSPE-PEG_2000_-OCH_3_), and CarboxyPEG(5000)-Distearoyl phosphatidylethanolamine (ammonium salt) (DSPE-PEG_5000_-COOH) were supplied by Nanocs (Natick, MA, USA). N-(3-dimethylaminopropyl)-N’ethyl-carbodiimide hydrochloride (EDC), 2-(N-morpholino) ethanesulfonic acid (MES), N-hydroxysulfosuccinimide (sulfo-NHS), glycine, the Phospholipid Assay kit, and the QuantiPro™BCA Assay kit were obtained from Sigma Aldrich (St. Louis, MO, USA). OX26 and the Pierce™F(ab′)_2_ Micro Preparation kit were purchased from Fisher Thermo Scientific (Waltham, MA, USA). Pilocarpine, methyl-scopolamine, diazepam, carbamazepine, tetraglycol, and polyethyleneglycol 600 were also obtained from Sigma Aldrich (St. Louis, MO, USA). Chloroform, *tert*-Butyl methyl ether (t-BME), HPLC-grade methanol (MeOH), and acetonitrile (ACN) were purchased from Merck (Arequipa, Peru). Ultrapure water was freshly obtained before use from a Merck Simplicity water purification system. All the other reagents were purchased from Sigma Aldrich (Arequipa, Peru).

### 2.2. Preparation of N-3-Methoxybenzyl-linoleamide-Loaded Conventional Liposomes (CL) and N-3-Methoxybenzyl-linoleamide-Loaded PEGylated Liposomes (PL)

Conventional and PEGylated liposomes were prepared by the thin film hydration method followed by sonication [45]. In summary, DMPC, either alone or combined with DSPE-PEG_2000_-OCH_3_ (Table 1), was weighed and dissolved in chloroform. N-3-methoxybenzyl-linoleamide was also added to the chloroform mixture alongside DMPC, maintaining its concentration at 5.0% (*w*/*w*) relative to the total lipids. The chloroform was then evaporated under reduced pressure using a rotary evaporator (Büchi R-100), creating a lipid film. This lipid film was hydrated at 40 °C with phosphate buffer solution (PBS) at pH 7.4, achieving a final phospholipid concentration of 10.0 μmol/mL. To create Multilamellar Vesicles (MLVs), the mixture underwent bath sonication above 40 °C for 10 min. After allowing the MLV suspension to rest for 10 min to resolve any structural defects, its size was adjusted through sonication at 40 °C. This process involved three cycles of 10 min each, using a 3 mm diameter probe sonicator (Intelligent Ultrasonic Processor, SJIA-650 W). The resulting Small Unilamellar Vesicles (SUVs) were then centrifuged at 10,000× *g* for 10 min to eliminate titanium particles shed by the probe and to precipitate non-encapsulated N-3-methoxybenzyl-linoleamide. Both formulations were subsequently stored in the dark at 4 °C to assess size stability. PEGylated liposomes to be used in the in vivo study were administered within 24 h after formulation.

### 2.3. Preparation of OX26 F(ab′)_2_ Fragments

The generation and purification of OX26 F(ab′)_2_ fragments were performed using the Pierce™F(ab′)_2_ Micro Preparation kit following the manufacturer’s instructions. In summary, 3.0 mL of the antibody, at a concentration of 1.0 mg/mL in 20 mM sodium acetate, pH 4.4, was concentrated to 0.5 mL using ultra 30 K centrifugal filters at 15,000× *g* at 4 °C for 5 min. The concentrated antibody solution was then added to equilibrated immobilized pepsin and incubated at 37 °C for 2.0 h with gentle shaking. The separation of F(ab′)_2_ fragments from undigested antibodies and Fc fragments was achieved via affinity chromatography using an immobilized protein A column and the manufacturer’s IgG elution buffer, pH 2.8. To eliminate residual small Fc fragments, centrifugation was conducted at 1000× *g* with ultra 30 K centrifugal filters.

### 2.4. Preparation of N-3-Methoxybenzyl-linoleamide-Loaded PEGylated OX26 F(ab′)_2_ Immunoliposomes (IL)

PEGylated immunoliposomes containing N-3-methoxybenzyl-linoleamide were prepared using the thin film hydration method followed by sonication [45]. The OX26 F(ab′)_2_ fragments were then conjugated to the distal carboxylic groups of the linker lipid DSPE-PEG_5000_-COOH through a carbodiimide coupling reaction [46]. In summary, DMPC, DSPE-PEG_2000_-OCH_3_, and DSPE-PEG_5000_-COOH (Table 2) were weighed and dissolved in chloroform. N-3-methoxybenzyl-linoleamide was added to the lipid mixture in chloroform at a concentration of 5.0% (*w*/*w*) relative to the final lipid concentration, 10 μmol/mL. The subsequent steps to produce SUV liposomes followed the same procedure as used for preparing conventional and PEGylated liposomes, with the exception that the dried lipid film was hydrated at 40 °C, using MES buffer at pH 5.2, as previously reported [41].

The carbodiimide reaction was carried out by mixing 50 mM EDC and 100 mM Sulfo-NHS with the liposomes in 100 mM MES buffer, pH 5.2. The mixture was incubated with gentle shaking for 15 min and subsequently dialyzed (100 kDa cut-off) against PBS, pH 7.4, at room temperature for 15 min to remove excess reagents. This dialysis step was repeated three additional times for a total of one hour. Following dialysis, the pH of activated liposomes was adjusted to 7.4 with a sodium hydroxide solution. Next, OX26 F(ab′)_2_ fragments were added to the activated liposomes, and the mixture was incubated at room temperature for 2.0 h, followed by overnight incubation at 4 °C. The reaction was terminated by adding 50 mM glycine and incubating for 30 min. The resulting immunoliposomes were purified with exclusion chromatography consisting of a glass column packed with Sephadex G-50 and PBS, pH 7.4. For the in vivo studies, immunoliposomes were concentrated using ultra 50 K centrifugal tubes. The immunoliposomes were then stored in the dark at 4 °C to assess size stability. Immunoliposomes to be used in the in vivo study were administered within 24 h after formulation. Blank immunoliposomes (lacking OX26 F(ab′)_2_ fragments) were produced following the same protocol.

### 2.5. Characterization of Liposomal Formulations

#### 2.5.1. Size and Zeta Potential

The size and polydispersity index (PDI) of liposomes were measured by dynamic light scattering using a particles analyzer, Zetasizer ZS90 (Malvern Instrument, Malvern, UK), at 25 °C and a scattering angle of 90°. The zeta potential was determined using the same device, which automatically adjusted the voltage. For all measurements, 100 μL of liposome suspensions were diluted with 900 μL of PBS 0.01 M, pH 7.4.

#### 2.5.2. Phospholipids Content

DMPC was quantified according to Phospholipid Assay kit, which determines the choline content using choline oxidase and a H_2_O_2_-specific dye. This results in a colorimetric product directly proportional to the phospholipid concentration in the sample. Absorbance was measured at 570 nm using a microplate reader (Biotek Synergy HTX, Santa Clara, CA, USA).

#### 2.5.3. N-3-Methoxybenzyl-linoleamide Loading Efficiency (DLE)

A 100 μL aliquot of each type of liposome formulation was lysed with 400 μL of acetonitrile. The mixture was centrifuged at 15,000× *g* for 10 min to precipitate acetonitrile-insoluble compounds. The resulting supernatant, containing N-3-methoxybenzyl-linoleamide, was diluted with methanol. The analytical procedure was carried out on a Waters Xevo G2-XS QTof liquid chromatography/mass spectrometer. N-3-methoxybenzyl-linoleamide and its internal standard, N-benzyl-palmitamide, were well-separated on an Acquity UPLC^®^ BEH C18 1.7 μm, 2.1 × 100 mm column at 35 °C. The mobile phase consisting of acetonitrile with formic acid (0.1%) and water with formic acid (0.1%) (95:5 *v*/*v*) was delivered in isocratic mode at 0.25 mL/min. The autosampler was set to 5 μL. The compounds were quantitated using positive electrospray ionization (ESI) with single ion monitoring (SIM) mode at *m*/*z* 400.3 for N-3-methoxybenzyl-linoleamide and *m*/*z* 346.6 for N-benzyl-palmitamide. Nitrogen was used as the nebulizing gas at 1.5 L/min. The collision energy was 10.0 V and the cone energy was 40.0 V.

Under these conditions, a calibration curve for N-3-methoxybenzyl-linoleamide was prepared using six standard concentrations: 12.5, 25, 50, 100, 200, and 400 ng/mL. The DLE (%) was calculated using the following formula: DLE (%) = (amount of drug in liposomes/amount of drug initially added) × 100.

#### 2.5.4. Quantification of Antibody-Liposome Coupling Efficiency

Aliquots of the eluates from the exclusion chromatography used in “Preparation of N-3-methoxybenzyl-linoleamide-loaded PEGylated OX26 F(ab′)_2_ immunoliposomes (IL)” were analyzed using the QuantiPro™BCA Assay kit. The procedure involves forming a Cu^2+^-protein complex under alkaline conditions, followed by the reduction of Cu^2+^ to Cu^1+^. The extent of the reduction is directly proportional to the protein content in the samples. Absorbance was measured at 562 nm and values were corrected using data obtained from blank immunoliposomes.

### 2.6. Animals

The anticonvulsant properties of free and liposomal N-3-methoxybenzyl-linoleamide were evaluated in male Sprague-Dawley rats housed at the Animal Facility of Universidad Andina del Cusco. The experiments adhered to the guidelines outlined in the Guide for the Care and Use of Laboratory Animals (Institute of Laboratory Animal Resources, National Research Council, National Academy of Sciences, Washington, DC, USA). Approval for the studies was granted by the Institutional Animal Care and Use Committee of Universidad Peruana Cayetano Heredia. Efforts were made to minimize animal distress. The rats were maintained under a 12 h light/dark cycle in a temperature-controlled environment, with unrestricted access to food and water. All animals acclimated for one week before the experiments and were seven weeks old, weighing between 250 and 300 g.

### 2.7. Study Design

Status epilepticus was induced in Sprague-Dawley rats through a single intraperitoneal injection of pilocarpine, following the protocol described by Turski et al. [47]. To mitigate the peripheral cholinomimetic effects of pilocarpine, the animals first received a subcutaneous injection of methyl-scopolamine at a dose of 1.0 mg/kg body weight. Thirty minutes later, pilocarpine was administered intraperitoneally at a dose of 350 mg/kg, which was selected based on the latency and mortality data from our pilot study. Status epilepticus was characterized as a phase of continuous seizures lasting at least 5 min or recurrent seizures occurring at intervals of less than one minute, indicating a persistent epileptiform state [48]. Once status epilepticus was induced, seizure activity and severity were assessed using Racine’s scale [49], which categorizes seizure progression as follows: 1 = seizure consisted of immobility and occasional facial clonus; 2 = head nodding; 3 = bilateral forelimb clonus; 4 = rearing; 5 = rearing and falling.

### 2.8. Experimental Procedure

The animals were randomly assigned to seven experimental groups (*n* = 6). One hour after entering status epilepticus, each group received a specific treatment administered as a single intravenous bolus through the caudal vein [50]. The dosages were determined based on our pilot study. Group 1: Vehicle of Diazepam, Carbamazepine, a synthetic macamide, which was a mixture of tetraglycol, polyethyleneglycol 600, and water (1:1:3). Group 2: Diazepam, at a dose of 4.0 mg/kg body weight. Group 3: Carbamazepine, at 25.0 mg/kg body weight dose. Groups 4 and 5: N-3-methoxybenzyl-linoleamide, at doses of 1.0 and 10.0 mg/kg body weight, respectively. Groups 6 and 7: PEGylated liposomes and immunoliposomes encapsulating N-3-methoxybenzyl-linoleamide at a dose of 1.0 mg/kg body weight. Their behavior was continuously observed and recorded for the following 48 h.

### 2.9. Statistical Analysis

Statistical analysis was conducted using a licensed version of SigmaStat 3.5 software. A normality test was applied to determine if data was normally distributed. Then, the Student’s *t*-test was used to assess the statistical significance between two groups, while one-way analysis of variance (ANOVA) followed by the Holm–Sidak post hoc test was applied for comparisons among multiple groups. A *p*-value of less than 0.05 was considered statistically significant.

### 2.10. Structures of the Macamide 3-MBL

The molecular structures and the electrostatic potential surface were obtained using Gaussian 16 [51] and GaussView v6 [52] software.

## 3. Results and Discussion

### Characterization of Liposomal Formulations

The size of drug delivery systems plays a crucial role in their uptake and permeability within the Central Nervous System (CNS). Nanoscale dimensions influence both pharmacokinetics and pharmacodynamics under physiological conditions (Figure 1) [53]. Nanosystems smaller than 10 nm are rapidly eliminated via kidney clearance or extravasation, while those larger than 200 nm are effectively filtered by the liver, spleen, and bone marrow. Therefore, a size range from 10 to 200 nm allows liposomes to remain in the bloodstream longer [54,55] and increase the time during which the nanosystem could be in contact with the blood–brain barrier [56]. In that sense, for the development of an appropriate immunoliposomal formulation encapsulating N-3-methoxybenzyl-linoleamide, conventional and PEGylated liposomes were previously designed and evaluated. DMPC was selected as the base phospholipid for the formulation instead of DPPC (as evaluated in our pilot study), either alone or with cholesterol, because DPPC produced conventional liposomes with a size exceeding 200.0 nm and an encapsulation efficiency of N-3-methoxybenzyl-linoleamide below 60.0%. In contrast, DMPC-based conventional liposomes demonstrated a smaller size, 79.69 ± 3.65 nm, and a higher encapsulation efficiency, 90.76 ± 1.40%, establishing a baseline structure for further optimization. Moreover, in PEGylated liposomes, the lack of cholesterol was remedied with the presence of DSPE-PEG_2000_-OCH_3_, which provides flexibility and stability and prolongs circulation time [57]. Due to the presence of DSPE-PEG_2000_-OCH_3_, a flexible, hydrophilic polymer with a length of about 12.5 nm [58], PEGylated liposomes were 14.27 nm larger than conventional liposomes. Immunoliposomes prior to OX26 F(ab′)_2_ coupling measured 105.4 ± 6.98 nm, reflecting a size increase of 25.71 and 11.44 nm related to conventional and PEGylated liposomes, respectively. This is due to the addition of DSPE-PEG_5000_-COOH, whose length is about 31.7 nm [58]. Following OX26 F(ab′)_2_ coupling, immunoliposomes reached 120.52 ± 9.46 nm, indicating the successful incorporation of the targeting ligand, which usually causes an enlargement between 7 and 20 nm [41,59]. The size of the liposomal formulations exhibits a systematic increase with each modification step. Nonetheless, all formulations maintained a polydispersity index below 0.30, ensuring acceptable size distribution and formulation uniformity [60].

Any significant change in the mean size of conventional and PEGylated immunoliposomes was registered while formulations were kept in PBS, in the dark, and at 4 °C for one week.

Given that neutral zeta potential falls within the range from −10.0 mV to +10.0 mV [61], the formulations exhibit a neutral zeta potential overall. However, immunoliposomes prior to OX26 F(ab′)_2_ coupling display the most negative value, −15.32 ± 1.37 mV, likely attributed to the presence of DSPE-PEG_5000_-COOH, whose carboxylic group would be deprotonated and negatively charged. The phospholipid content is similar for conventional liposomes, long-circulating liposomes, and immunoliposomes before OX26 F(ab′)_2_ coupling, with values of 92.46 ± 3.12%, 93.24 ± 18.84%, and 91.51 ± 2.80%, respectively. The dialysis step may have caused a decrease in phospholipid content in immunoliposomes after OX26 F(ab′)_2_ coupling, as this formulation reached only 86.40 ± 4.35%. Similarly, the loading efficiency of N-3-methoxybenzyl-linoleamide highlights the structural and compositional changes that occur during liposomal modification, with the most challenging step being the inclusion of OX26 F(ab′)_2_. This may explain the decrease of N-3-methoxybenzyl-linoleamide loading to 79.42 ± 6.68% after such an inclusion.

While the coupling efficiency of the OX26 antibody or its F(ab′)_2_ fragments to PEGylated liposomes varies depending on the specific conjugation method and conditions employed, herein, the OX26 F(ab′)_2_ coupling efficiency for the immunoliposomes was 33.75 ± 3.8%, higher than previous studies [41]. Status epilepticus was induced via intraperitoneal administration of pilocarpine at a dose of 350 mg/kg. This dosage was selected based on our pilot study, considering latency and mortality rates, and is consistent with prior research [62]. Following the pilocarpine injection, experimental animals exhibited intermittent seizures for one hour, progressing to status epilepticus. This condition was characterized by mouth and facial clonus, head nodding, forelimb clonus, rearing on hind legs, and falling, as described by Racine’s scale [49].

Two controls were used in this study to evaluate the anticonvulsant effect. The first was diazepam, administered at a dose of 4.0 mg/kg, a primary pharmacotherapy for halting status epilepticus [63,64]. Diazepam was considered to have a 100% anticonvulsant effect between 0.25 and 3.0 h post-treatment. The second control was carbamazepine, administered at a dose of 25.0 mg/kg. Carbamazepine is one of the older antiepileptic drugs approved by the FDA [65,66,67]. As shown in Figure 2, carbamezepine reduced the signs of status epilepticus between 80.73 and 87.40% compared to diazepam in the first 3.0 h post-treatment (Table 3).

As expected, N-3-methoxybenzyl-linoleamide at 1.0 mg/kg showed a mild anticonvulsant effect compared to diazepam and carbamazepine. However, at a dose of 10.0 mg/kg, this macamide relieved the number and severity of seizures, showing an effect similar to diazepam and slightly superior to carbamazepine, from 0.5 to 2.0 h post-treatment.

Although conventional liposomes loaded with N-3-methoxybenzyl-linoleamide were formulated in this study, they were not used for the in vivo assays due to limitations such as rapid clearance from the bloodstream and uptake by the reticuloendothelial system (RES). This process, primarily driven by plasma protein deposition, results in their swift removal from circulation and their accumulation in organs like the liver and spleen [68], as well as the limited brain uptake of liposome-encapsulated drugs [41].

At all different time points, PEGylated liposomes encapsulating N-3-methoxybenzyl-linoleamide exhibited a significantly lower anticonvulsant response compared to diazepam, carbamazepine, and free N-3-methoxybenzyl-linoleamide at 10.0 mg/kg. This may be attributed to the limited penetration through the blood–brain barrier, despite the longer half-life of the liposomal cargo. Previous in vivo studies have shown that while some PEGylated liposomal drugs achieve efficacy comparable to the free form [69,70,71], others demonstrate greater efficacy than the free form in treating central nervous system diseases [72,73]. Herein, PEGylated liposomes displayed a higher anticonvulsant effect than free N-3-methoxybenzyl-linoleamide at 1.0 mg/kg only at 1.0 and 3.0 h post-administration.

Active targeting liposomes, specifically immunoliposomes loaded with N-3-methoxy-benzyl-linoleamide, formulated in this study, showed a significantly lower anticonvulsant effect compared to diazepam but was significantly similar to that obtained with carbamazepine up to 2.0 h post-treatment.

The direct coupling of ligands to the surface of PEGylated liposomes has been explored for active targeting, but the PEG chains exert a strong steric shielding effect, hindering the interaction between the anchored ligand and its receptor [74]. Consequently, PEG is primarily used as a spacer, with targeting ligands attached to the termini of the PEG chains to enhance ligand accessibility and flexibility. In our study, DSPE-PEG_5000_-COOH was used as a spacer to overexpose OX26 F(ab′)_2_ fragments. In addition, DSPE-PEG_2000_-COOH might achieve longer circulation and enhanced BBB transcytosis compared to DSPE-PEG_2000_-COOH [75].

The monoclonal antibody OX26 is a highly effective tool for brain targeting, crossing the BBB via receptor-mediated transcytosis [76,77]. It binds specifically to BCECs due to the high expression of TfR on their luminal surface [78]. Using OX26 F(ab′)_2_ fragments instead of the full antibody was based on their smaller size, which generates smaller liposomes and potentially enhances BBB penetration [79,80,81]. Additionally, full antibodies can interact with Fc receptors on various peripheral cells, leading to off-target effects. In contrast, OX26 F(ab′)_2_ fragments lack the Fc region, minimizing such interactions and avoiding immune responses [79,82].

From 3.0 h onwards, no seizures were observed in the groups receiving pharmacological treatments. Therefore, the anticonvulsant effects of PEGylated liposomes and immunoliposomes could not be evaluated beyond this time point. Thus, a prolonged pharmacodynamic study should not only include status epilepticus but also the latent phase and seizures during the chronic phase.

Pharmacokinetics, the science that describes the disposition of drug candidates in the body, including absorption, distribution, metabolism, and excretion, as well as biodistribution [83,84], which refers to the reversible transfer of a drug candidate from the blood to various extravascular spaces, are primary considerations in the drug development process [85,86]. Therefore, future studies should assess the pharmacokinetics and biodistribution of 3-MBL and the formulations initially proposed in this manuscript.

PEG has generally been recognized for its non-toxicity, ability to extend the half-life, and role in enhancing the biological activity of associated molecules [87,88,89]. However, in recent years, anti-PEG antibodies have been reported [90]. For example, studies have shown that lipid nanoparticles could induce anti-PEG antibodies [91], which may, in turn, alter and reduce the therapeutic efficacy of PEG-associated molecules [92]. Therefore, special attention should be given to developing the active targeting nanocarrier proposed in this study. Since molecular weight [93,94] and terminal groups [95] can trigger different immune responses, some protocols suggest dosage adjustments or the concurrent use of immunomodulators [96].

Although immunoliposomes are one of the most promising strategies for reaching specific therapeutic targets [97,98], their translation into human therapies requires careful design. Factors such as prolonged observation times, pharmacokinetic studies, biodistribution, toxicity, immunology, possible dosing regimens, improved drug stability, etc., are necessary elements in its design [99]. However, this encapsulation has been found to increase biocompatibility, biodegradability, and non-immunogenicity, considerably decreasing its toxicity. Although it has been observed that macamides have low or no toxicity [100], the use of immunoliposomes could prolong their half-life in the body, allowing therapeutic targets to be reached and increasing their efficacy in the treatment, not only of epilepsy but of various diseases.

## 4. Conclusions

In summary, this study has formulated a nanoliposomal formulation to deliver N-3-methoxybenzyl-linoleamide and improve its anticonvulsant effects. This strategy proposes various advantages such as improved solubility, enhanced circulation time through PEGylation, enhanced nervous system delivery trough the conjugation of OX26 F(ab′)_2_ fragments, decreased therapeutic dose, and thereby reduced peripheric distribution. These benefits allow the anticonvulsant effect of N-3-methoxybenzyl-linoleamide encapsulated in immunoliposomes to be compared to carbamazepine, an efficient and traditional antiepileptic.

## Figures and Tables

**Figure 1 neurolint-17-00038-f001:**
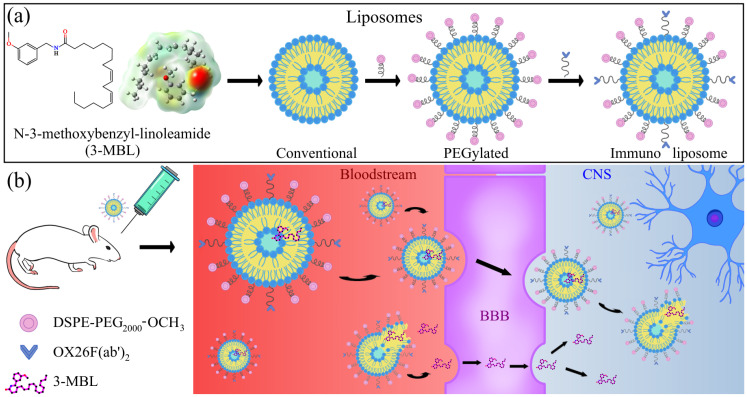
Schematic representation of the use of liposomes as carriers of macamide 3-MBL. (**a**) Macamide and liposomes used in in vivo tests. The colors on the surface of the calculated electrostatic potential for 3-MBL indicate regions of high electron density (red) and low electron density (blue). Neutral zones are represented in white. (**b**) Possible mechanisms of action of immunoliposomes in the human brain.

**Figure 2 neurolint-17-00038-f002:**
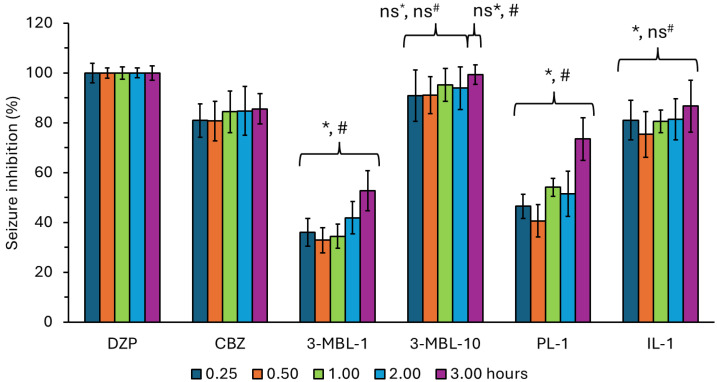
Seizure inhibition (%) on pilocarpine-induced status epilepticus after the administration of diazepam (DZP), carbamazepine (CBZ), N-3-methoxybenzyl-linoleamide at 1.0 mg/kg (3-MBL-1), N-3-methoxybenzyl-linoleamide at 10.0 mg/kg (3-MBL-10), PEGylated liposomes loaded with N-3-methoxybenzyl-linoleamide at 1.0 mg/kg (PL-1), and PEGylated OX26 F(ab′)2 immunoliposomes loaded with N-3-methoxybenzyl-linoleamide at 1.0 mg/kg (IL-1). The percentage of seizure inhibition is expressed as the mean ± S.D. Bars represent the standard deviation. *n* = 6. * *ANOVA test* = Significantly different to the response displayed by the group that received Diazepam at 4.0 mg/kg. # *ANOVA test* = significantly different to the response displayed by the group that received carbamazepine at 25.0 mg/kg. *ns** = not significantly different to the response displayed by the group that received diazepam at 4.0 mg/kg. *ns^#^* = not significantly different to the response displayed by the group that received Carbamazepine at 25.0 mg/kg.

**Table 1 neurolint-17-00038-t001:** Composition, size, polydispersity index (PDI), and zeta potential of liposomal formulations (*n* = 3).

Formulation	Composition	Molar Ratio	Mean Particle Diameter (nm)	PDI	Z-Potential (eV)
Conventional liposomes	DMPC	100	79.69 ± 3.65	0.14 ± 0.02	−1.74 ± 0.31
PEGylated liposomes	DMPC:DSPE-PEG_2000_-OCH_3_	100:7.5	93.96 ± 4.00	0.16 ± 0.01	−3.17 ± 0.69
Immunoliposomes before	DMPC:DSPE-PEG_2000_-OCH_3_:	100:6.25:1.25	105.40 ± 6.98	0.19 ± 0.03	−15.32 ± 1.3
OX26 F(ab′)_2_ coupling	DSPE-PEG_5000_- COOH				
Immunoliposomes after	DMPC:DSPE-PEG_2000_-OCH_3_:	100:6.25:1.25	120.52 ± 9.46	0.23 ± 0.03	−8.57 ± 0.80
OX26 F(ab′)_2_ coupling	DSPE-PEG_5000_- COOH				

**Table 2 neurolint-17-00038-t002:** Composition, phospholipid content, and N-3-methoxybenzyl-linoleamide loading efficiency of liposomal formulations (*n* = 3).

Formulation	Composition	Molar Ratio	Phospholipids Content (%)	Loading Efficiency (%)
Conventional liposomes	DMPC	100	92.46 ± 3.12	90.76 ± 1.40
PEGylated liposomes	DMPC:DSPE-PEG_2000_-OCH_3_	100:7.5	93.24 ± 1.84	87.37 ± 2.43
Immunoliposomes before	DMPC:DSPE-PEG_2000_-OCH_3_:	100:6.25:1.25	91.51 ± 2.80	86.25 ± 0.70
OX26 F(ab′)_2_ coupling	DSPE-PEG_5000_- COOH			
Immunoliposomes after	DMPC:DSPE-PEG_2000_-OCH_3_:	100:6.25:1.25	86.40 ± 4.35	79.42 ± 6.68
OX26 F(ab′)_2_ coupling	DSPE-PEG_5000_- COOH			

**Table 3 neurolint-17-00038-t003:** Seizure inhibition (%) on pilocarpine-induced status epilepticus after the administration of diazepam (DZP), carbamazepine (CBZ), N-3-methoxybenzyl-linoleamide at 1.0 mg/kg (3-MBL-1), N-3-methoxybenzyl-linoleamide at 10.0 mg/kg (3-MBL-10), PEGylated liposomes loaded with 3-MBL at 1.0 mg/kg (PL-1), and PEGylated OX26 F(ab′)_2_ immunoliposomes loaded with 3-MBL at 1.0 mg/kg (IL-1). *n* = 6.

Drug	Time Post Administration of Treatments (Hours)											
	0.25		0.50		1.00		2.00		3.00		6.00	
	Mean	SD	Mean	SD	Mean	SD	Mean	SD	Mean	SD	Mean	SD
DZP	100.00	3.88	100.00	2.06	100.00	2.49	100.00	2.00	100.00	2.87	100.00	0.00
CBZ	80.91	6.64	80.73	7.96	84.43	8.33	84.75	9.76	85.53	6.09	87.40	2.41
3-MBL-1	36.10	5.57	32.87	5.04	34.45	4.83	41.86	6.55	52.72	7.94	69.79	2.42
3-MBL-10	90.93	10.31	91.16	7.44	95.23	6.59	93.89	8.59	99.28	3.89	97.47	6.21
PL-1	46.48	4.84	40.57	6.49	54.12	3.58	51.50	9.07	75.53	8.57	76.34	5.98
IL-1	81.05	8.00	75.32	9.18	80.55	4.57	81.40	8.30	86.66	10.42	97.47	6.21

## Data Availability

The data presented and analyzed in this study are available from the corresponding author upon reasonable request.

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
