# Peer review of "Using Immunoliposomes as Carriers to Enhance the Therapeutic Effectiveness of Macamide N-3-Methoxybenzyl-Linoleamide"

_2035-8377, 2025, doi:10.3390/neurolint17030038_

Round 1
Reviewer 1 Report
Comments and Suggestions for Authors
Using Immunoliposomes as Carriers to Enhance the Therapeutic Effectiveness of Macamide N-3 Methoxybenzyl-Linoleamide
The article presents a study on the use of immunoliposomes to enhance the anticonvulsant properties of synthetic macamide N-3 methoxybenzyl-linoleamide (3-MBL), which is analog of natural secondary metabolite derived from Maca. The study integrates nanomedicine and targeted drug delivery, aiming to improve therapeutic outcomes for epilepsy patients while minimizing systemic side effects. The authors employ PEGylated liposomes conjugated with OX26 F(ab’)2 fragments to facilitate brain targeting via transferrin receptor-mediated transcytosis.
A few strengths of the manuscript can be noted. First, innovative approach (the study successfully incorporates targeted nanocarrier technology to improve the bioavailability and efficacy of 3-MBL, addressing a major challenge in CNS drug delivery); second, comprehensive methodology and detailed physicochemical characterization (the experimental design includes a well-structured characterization of liposomes (particle size, zeta potential, phospholipid content, and drug loading efficiency), evaluation of pharmacokinetics, and in vivo testing in a pilocarpine-induced rat epilepsy model; and third, comparison with standard treatments (comparison with FDA-approved anticonvulsants such as carbamazepine and diazepam, the study provides a relevant clinical context for the efficacy of the proposed formulation).
But the study also has serious limitations, as:
- Limited Pharmacokinetic and Biodistribution Data: The study lacks a detailed analysis of the pharmacokinetics and biodistribution of the immunoliposomes compared to free 3-MBL. Such data would provide deeper insights into the systemic circulation time and precise brain uptake.
- Short-Term Evaluation of Efficacy: The study observes seizure suppression for only 3 hours post-treatment. A longer observation period would be necessary to evaluate sustained efficacy and potential delayed side effects.
Recommendations for corrections to be introduced in the text:
- While the authors suggest that 3-MBL exerts its anticonvulsant effects via FAAH inhibition and endocannabinoid modulation, they do not provide experimental evidence supporting this mechanism. Introduction of data from studies involving receptor binding assays or molecular docking could strengthen these claims.
- The study does not address potential immune responses to PEGylated liposomes, which could impact long-term treatment viability. Discuss potential Immunogenicity of PEGylated liposomes.
- Add discussion about the molecular mechanism of 3-MBL’s anticonvulsant effects.
- The study is limited to a rodent model of epilepsy, and no discussion is provided on potential challenges in translating these findings to human clinical trials, such as dose scaling and metabolic differences. Discuss potential translation to human application!
Reviewer 2 Report
Comments and Suggestions for Authors
Line 76: It is suggested to correct: Blood-Brain Barrier (BBB)
Line 89: As I already pointed out before, it is only necessary to write “BBB”
Line 90: It is suggested to correct: Brain Capillary Endothelial Cells (BCECs), review the entire manuscript and adjust the use of capital letters where necessary.
Line 105 and 106: Check and correct the long name of the chemical compound, using a shorter name for the compound, Check in PubChem.
In the compounds used in the experiments, in addition to adding the brand and the country, the state is necessary.
Line 122: mentions chloroform but does not add it in the materials section.
In section 2.9. It is not mentioned if the data showed a normal distribution, homogeneity in the variances, to determine whether to perform an ANOVA or Students' t-test, it will also be important to mention that the statistical program is licensed.
Line 269: Check that 90.76± 1.40% use the symbol “±” separated between the numerical values, but in tables 1 and 2 they do not do so, the criteria must be homogenized throughout the manuscript.
It does not mention which program or version of the same used to make figure 1.
Reference 2, the year is not in bold.
17 references are from the last 5 years, which represents 27% of the 62, so at least 51% is necessary.
